# Wildfire Smoke Exposure during Pregnancy: A Review of Potential Mechanisms of Placental Toxicity, Impact on Obstetric Outcomes, and Strategies to Reduce Exposure

**DOI:** 10.3390/ijerph192113727

**Published:** 2022-10-22

**Authors:** Emilia Basilio, Rebecca Chen, Anna Claire Fernandez, Amy M. Padula, Joshua F. Robinson, Stephanie L. Gaw

**Affiliations:** 1Division of Maternal-Fetal Medicine, Department of Obstetrics, Gynecology & Reproductive Sciences, University of California, San Francisco, CA 94143, USA; 2School of Public Health, University of California, Berkeley, CA 94704, USA; 3Center for Reproductive Sciences, Department of Obstetrics, Gynecology & Reproductive Sciences, University of California, San Francisco, CA 94143, USA

**Keywords:** air pollution, pregnancy, wildfire, smoke, fire, placenta, toxicology, perinatal outcomes, climate change, pregnancy, pregnant, obstetrics, preterm birth, fetal growth, birth weight, reproduction, inflammation, epigenetics, oxidative stress, endocrine, metabolism, hormone, vascular, vasculogenesis, hypertension, PM_2.5_, PM_10_, inflammation

## Abstract

Climate change is accelerating the intensity and frequency of wildfires globally. Understanding how wildfire smoke (WS) may lead to adverse pregnancy outcomes and alterations in placental function via biological mechanisms is critical to mitigate the harms of exposure. We aim to review the literature surrounding WS, placental biology, biological mechanisms underlying adverse pregnancy outcomes as well as interventions and strategies to avoid WS exposure in pregnancy. This review includes epidemiologic and experimental laboratory-based studies of WS, air pollution, particulate matter (PM), and other chemicals related to combustion in relation to obstetric outcomes and placental biology. We summarized the available clinical, animal, and placental studies with WS and other combustion products such as tobacco, diesel, and wood smoke. Additionally, we reviewed current recommendations for prevention of WS exposure. We found that there is limited data specific to WS; however, studies on air pollution and other combustion sources suggest a link to inflammation, oxidative stress, endocrine disruption, DNA damage, telomere shortening, epigenetic changes, as well as metabolic, vascular, and endothelial dysregulation in the maternal-fetal unit. These alterations in placental biology contribute to adverse obstetric outcomes that disproportionally affect the most vulnerable. Limiting time outdoors, wearing N95 respirator face masks and using high quality indoor air filters during wildfire events reduces exposure to related environmental exposures and may mitigate morbidities attributable to WS.

## 1. Introduction

More than 73,000 wildfires burn an average of about 7 million acres of private, state, and federal land in the United States (US) each year. Due to drought, extreme heat, and reduced snowpack, recent wildfire seasons were unprecedented in severity and intensity [1]. The Natural Resources Defense Council estimates that 212 million people in the US lived in counties affected by wildfire conditions in 2011 and that wildfire smoke (WS) has accounted for up to 25% of PM_2.5_ (particulate matter with diameter < 2.5 μm) in recent years across the US and up to half in some Western regions [2]. In 2020, about 70% of the population in California experienced over 100 days of unhealthy air quality as specified by PM_2.5_ levels. The smoke of Western wildfires also spread to the East Coast in the summer of 2021, triggering restrictions on outdoor activities due to poor air quality, reflecting how smoke can disperse far from the source affecting areas downwind of burn sites [3]. Wildfires can temporarily increase air pollutant levels over thousands of square miles [4,5,6,7,8]. Global wildfire events have increased significantly in recent years, including in Canada, Australia, Europe, and Asia.

Air pollution has negative implications on pregnancy outcomes; thus, air pollution that results from WS is hypothesized to contribute to pregnancy morbidity as well [9,10,11]. Since components of WS can reach the placenta, they may adversely alter function leading to toxicity and adverse pregnancy outcomes.

Here, we review epidemiological and toxicological studies investigating the relationship between WS and pregnancy outcomes, especially in the placenta. When data specific to WS was unavailable, we included data from studies on general air pollution, particularly for components commonly found in WS, such as PM_2.5_. We highlight study designs, models, and biomarkers and mechanistic pathways that are applicable for future studies. Finally, we discuss interventions currently recommended to reduce exposure in pregnancy that that may impact individual-level exposures and are critical to consider in epidemiologic and translational studies of WS exposure.

### 1.1. The Complexity of Wildfire Smoke Composition 

Due to the burning of biomass and manmade structures, WS contains a mixture of particles and chemicals including high levels of carbon monoxide, carbon dioxide, and particulate matter (PM) [12]. PM generated by wildfires may be more toxic than non-fire period ambient PM, on an equal-mass basis, collected in the same region due to atmospheric photochemistry forming secondary pollutants [12,13].

WS not only increases the concentrations of PM_2.5_, but may also alter the composition of ambient PM_2.5_ [14,15]. PM_2.5_ composition can vary by region and season, and different PM_2.5_ compositions may impose different effects on human health [16,17,18,19]. Liu and colleagues assessed the population-based exposure to wildfire-related PM_2.5_ species from 2004–2009 in 51 Western US counties across six ecoregions [20]. By integrating monitor measurements on 29 PM_2.5_ species and data on “smoke waves”, they found that smoke waves were associated with an increase in the fraction of organic carbon, elemental carbon, and decreases in fractions of sulfate and crustal species of total PM_2.5_. A “smoke wave” was defined as ≥2 consecutive days with high wildfire-specific PM_2.5_, to describe episodes of high air pollution from wildfires [15].

Notably, developmentally toxic chemicals such as heavy metals (e.g., lead, cadmium, mercury) and specific polycyclic aromatic hydrocarbons (PAHs), especially those of low molecular weight (e.g., naphthalene, phenanthrene), are found at higher levels than typical ambient air pollution sources [21]. For example, ambient lead levels were 50 times higher than normal in major California cities afflicted by smoke from the Camp Fire in 2018 (the deadliest wildfire in California’s history as of this publication), stemming from the redistribution of soil contaminated with leaded gasoline from the 1960–80s [22,23]. Wildfires contribute to the emission of 1,900,000 kg per year of lead [24]. Wildfires are also a significant source of human exposure to diverse PAHs (over 100 known), including 16 identified as high-priority pollutants by the United States Environmental Protection Agency due their toxic effects (e.g., carcinogenesis, immune, developmental, and reproductive toxicity) [25]. WS also contains a mixture of other toxic chemicals (e.g., flame retardants, industrial solvents) that may negatively affect human health due to combustion of homes, furniture, and other materials.

### 1.2. Wildfire Smoke, PM_2.5_, and Impact on Perinatal Outcomes 

The adverse effects of smoke from maternal tobacco use, secondhand smoke, wood-burning cookstoves, and ambient air pollution are well-known, but the effects of WS on pregnancy are largely unknown. Pregnant individuals are especially susceptible to the adverse health effects of air pollutant exposure due to the increased respiration and cardiovascular output in pregnancy [26]. PM_2.5_ penetrate the lung and bloodstream more easily than larger sized PM [27]. Furthermore, recent fires occur at the wildland-urban interface leading to structure burning, which results in toxic air pollutants. To our knowledge, there have only been ten epidemiologic studies investigating the effects of WS on pregnancy outcomes, most of which have focused on preterm birth, birth weight, and one on preeclampsia (Table 1).

#### 1.2.1. Preterm Birth 

Preterm birth (PTB), birth at gestational age less than 37 weeks, is associated with a wide variety of comorbidities that threaten neonatal health, including chronic lung disease, developmental delays, growth reduction, as well as hearing and visual impairments [28]. Complications of severely preterm term birth are directly correlated with the gestational age at which neonates are born [28], therefore, elucidation of risk factors for PTB bears direct implications on neonatal survival rates. 

A study in Colorado found exposure to wildfire PM_2.5_ during pregnancy and during the second trimester between 2007–2015 was associated with higher risk of preterm birth, OR = 1.076 (95% CI: 1.016–1.139) and 1.132 (95% CI: 1.088–1.178), respectively) [29]. In California from 2006–2012, medium WS PM_2.5_ (5–10 μg/m^3^) and high WS PM_2.5_ (>10 μg/m^3^) smoke-days were associated with higher preterm birth risk (0.95% (95% CI: 0.47–1.42%) and 0.82% (95% CI: 0.41–1.24%), respectively) [30]. Their analysis estimated that 6974 excess preterm births were attributable to WS exposure out of a total of over 3 million births [30]. The current body of literature suggests an association between WS exposure and preterm birth, particularly when the exposure occurs in mid pregnancy [29]. 

#### 1.2.2. Fetal Growth

Birth weight serves as an important population-level predictor of infant mortality and morbidity with impact on adverse health outcomes later in life. A recent review on the topic demonstrated positive associations between low birth weight and future risk of cardiovascular disease, cancer, respiratory illness and chronic metabolic conditions [31]. On the other hand, neonates born large for gestational age–above the 90^th^ percentile for weight–are at risk of trauma secondary to complicated vaginal deliveries and conditions like hypoglycemia, independent of whether gestational diabetes was present during pregnancy [32]. Birth weight has been suggested to influence short term and long term neonatal outcomes such as cardiovascular diseases, diabetes, hypertension and stroke later in life [33]. 

Wildfires are hypothesized to influence fetal growth in complex ways, and researchers have yet to reach consensus on the exact impact of wildfire specific PM_2.5_ exposure on neonatal birth weight. Of the 7 studies on WS and birth weight, 6 found an association with decreased birth weight [29,32,34,35,36,37,38]. These wildfire-specific studies are summarized in Table 1. For example, exposure to WS PM_2.5_ in Colorado during the first trimester was associated with decreased birth weight by −5.7 g/(µg/m^3^) (95% CI: −11.1, −0.4) [29]. Pregnancy during the 2003 Southern California wildfires was associated with reduced average birth weight by 7.0 g (95% CI: −11.8, −2.2) when the wildfire occurred in the third trimester, 9.7 g less (95% CI: −14.5, −4.8) when it happened in the second, and 3.4 g lower (95% CI: −7.2, −0.6) in the first trimester [35]. On the contrary, an Australian study found that increased exposure to PM_2.5_ in WS correlated with increased birth weight (on average 197 g heavier in severely affected areas) when compared to pregnancies moderately affected by wildfires [32]. Interestingly, these findings were sex-specific—only males born to exposed mothers were born large for gestational age [32].

**Table 1 ijerph-19-13727-t001:** Summary of population-based studies in pregnancy included in our review. For each study, we provide the citation, time frame, location, study design, study population, and sample size. We also provide a description of the exposure. Finally, we indicated the primary and secondary outcomes observed, if applicable. Citations are listed in alphabetical order.

Citation	Type of Exposure	Time	Location	Study Design	Study Population	Sample Size (*n*)	Description of Exposure	Primary Outcome(s)	Secondary Outcome(s)
Abdo (2019) [29]	Wildfire smoke	2007–2015	Colorado, USA	Retrospective cohort	Singleton births with estimated gestational age between 30 and 42 weeks	446,961	Wildfire smoke PM_2.5_ and non-smoke PM_2.5_ linked by maternal residence ZIP code; method combined NOAA’s satellite imagery-based HMS to determine daily smoke plume extent with spatial interpolation of ground-based PM_2.5_ monitor values from US EPA AQS	PTB, BW	NICU admission, gestational diabetes, gestational hypertension, assisted ventilation at
Assibey-Mensah (2020) [39]	Wood smoke and traffic particle pollution	2009–2013	New York, USA	Retrospective cohort	Birth certificate data with gestation age estimated between 24 and 42 weeks	20,596	PM_2.5_, black carbon, Delta-C (wood smoke marker) concentrations, temperature, and relative humidity measured hourly, gestational month-specific concentrations of each at maternal residential address estimated using land-use regression model	Preeclampsia, any type (early- and late-onset)	
Breton (2011) [38]	Wildfire smoke	2003–2004	California, USA	Retrospective cohort	Pregnant women living in southern California	Not reported	Wildfire PM_2.5_ and ambient PM_2.5_ assigned exposures from maternal addresses geocoded for week of wildfire, exposure estimates derived in a GIS framework, MODIS satellite imaging used to obtain smoke information	BW	SGA, PTB
Heft-Neal (2022) [30]	Wildfire smoke	2006–2012	California, USA	Retrospective cohort	Singleton births in California with estimated gestation age between 23 and 41 weeks	3,002,014	Wildfire smoke plume extent assembled from NOAA’s satellite imagery-based HMS, high-resolution, temporally and spatially resolved gridded estimates of surface PM_2.5_ developed using machine learning algorithms to incorporate ground monitor data, chemical transport model predictions, and satellite observations; exposures linked to maternal address ZIP code	PTB	PTB subtype and severity (<28 weeks, <32 weeks)
Holstius (2012) [35]	Wildfire smoke	2001–2005	California, USA	Time-series	Birth records from the California Automated Vital Statistics System for infants delivered at term with BW between 1–6 kg	886,034	Temporally defined smoke exposure from MODIS satellite imagery, sensitivity analysis using maternal residence census tracts closer to monitors with average PM_10_ of <40 µg/m^3^ classified as low exposure, >40 µg/m^3^ classified as high exposure.	BW	
Jayachandran (2009) [40]	Wildfire air pollution	1997	Sumatra, Indonesia	Ecological	Birth cohorts conceived before or after wildfire events	3751	TOMS aerosol index	Fetal loss, infant mortality	
McCoy (2016) [37]	Wildfire smoke	2002–2013	Colorado, USA	Retrospective cohort	Live births in Colorado with maternal home address within 20 miles of a fire burn and smoke plume	7398	Proximity of self-reported maternal residence to wild-fire smoke using fine-scaled spatial dataset of plumes in GIS from satellite images of 28 wildfires in Colorado	BW, GA	
O’Donnell (2013) [34]	Wildfire event	2009	Victoria, Australia	Retrospective cohort	All births registered in Victoria	287,688	Proximity of maternal residence to Black Saturday wildfires	BW, PTB, changes to sex-ratio	
O’Donnell (2015) [32]	Wildfire event	2000–2010	Canberra, Australia	Retrospective cohort	All births registered in Canberra	48,408	Proximity of maternal residence to wildfire	BW, GA	
Prass (2012) [36]	Forest fire event	2001–2006	Porto Velho, Brazil	Cross- sectional	Singleton live births	22,012	Heat spots (all forest fires in Amazon region from 2001–2006) compared to time periods with lowest numbers of heat spots using NOAA satellite images	BW	

Abbreviations: GA = gestational age; GIS = Geographic Information System; MO = Moderate Resolution Imaging Spectroradiometer; NOAA = National Oceanic and Atmospheric Ad-ministration; NICU = neonatal intensive care unit; LBW = low birth weight; PM = particulate matter; PTB = preterm birth; SGA = small for gestational age; TOMS = Total Ozone Mapping Spectrometer.

The cause of the discrepancy in findings is unknown. However, they may be attributable to the methodologic differences in measurement of exposures. The quantification of exposure is challenging to capture, with some studies implementing spatial and/or temporal measurements paired with satellite imagery and geocoding of maternal residences. The use of stationary monitors in the studies that quantified air quality during wildfire events provide important insights at a population scale, however, they cannot capture individual differential exposures due to indoor versus outdoor air quality variation, occupation, and any behavioral changes implemented to reduce exposure. Additionally, there is altered composition and toxicity of WS from different events depending on the type biomass that is burning, distance from wildfire event, temperature, and age of particles [41]. 

### 1.3. Anatomy of the Human Maternal-Fetal Interface and Particulate Matter Deposition 

The placental chorionic villus is the critical barrier and exchange interface between the mother and fetus (Figure 1). Villous cytotrophoblasts (CTBs) differentiate into syncytiotrophoblasts (STBs) and extravillous CTBs. Multinucleated syncytiotrophoblasts (STBs) are formed by the fusion of villous CTBs to line the outer layer of the floating villus, whose branches literally “float” in maternal blood of the intervillous space [42,43]. The STB layer functions as the first physical, metabolic, and immunologic barrier to the fetal circulation, and is a significant source of hormone and cytokine production. Access to the fetal circulation is also regulated by a secondary layer of CTB attached to a basement membrane, and the stromal core which contains fibroblasts, fetal macrophages (Hofbauer cells), mesenchymal stem cells, and fetal endothelium [44]. Villous CTBs form columns of non-polarized cells that attach to and invade the uterine wall, giving rise to extravillous CTBs that interact closely with maternal decidual, myometrial, and immune cells to form anchoring villi [44]. Extravillous CTBs also interface with maternal endothelium, creating hybrid spiral arteries that divert uterine blood flow to the placenta. 

Numerous molecular cues, including gene pathways that mediate adhesion, migration, and cell–cell communication, underlie the multiple functional capabilities of trophoblasts. Altered or impaired placental function is significantly linked with common pregnancy complications, lower birth weights, developmental delay, and many diseases which manifest later in life [45,46,47,48].

The placenta serves multiple critical functions for embryonic/fetal growth and acts as a metabolic barrier to some environmental chemicals. The extent to which the placenta accumulates air pollutants was recently assessed by Bové and colleagues [49]. They postulated that black carbon derived from air pollution is transferred to maternal lung tissue, enters the systemic circulation, and translocates into placental tissue [49]. In their study, black carbon load was assessed in placental villous tissue using two-photon microscopy in the trophoblast cell layer (the outermost cellular layer in contact with maternal blood) [49,50]. These findings establish that the placental barrier is not impenetrable to particulate matter from air pollution exposures, although the extent of “transmission” to the fetus remains unknown. Their findings are consistent with a recent in vivo study of human first trimester trophoblast cell line HTR-8 that was exposed to wood smoke [51]. When exposed to wood smoke, particulate matter was identified inside exposed cells and localized to the mitochondria and endoplasmic reticulum using transmission electron microscopy [51]. Importantly, there are not yet investigations of human placental tissue describing particulate matter accumulation collected in pregnancies exposed to wildfire events. Notably, several chemical components commonly found in WS (e.g., heavy metal, PAHs) cause developmental toxicity and are known to accumulate in the placenta/fetus. 

### 1.4. Biologic Mechanisms of Damage from Wildfire, Air Pollution and Inhaled Toxins

WS likely impacts multiple biological mechanisms important for sustaining a healthy pregnancy. Previous reports on the relationship between air pollution and PM exposure with human health have pointed to inflammation, oxidative stress, endocrine, and cellular dysfunction as underlying mechanisms for poor outcomes [52,53,54,55,56]. Fine particles, free radicals, and reactive oxygen species formed through redox reactions between the particles themselves and lung tissue are likely involved in activating inflammatory cells releasing oxidative meditators, or it is possible that small particles that translocate from the alveolar membrane into the bloodstream interact with vascular endothelium cells via stimulation of enzymes [57]. As mentioned earlier in this paper, WS has various toxins such as heavy metals as well as PAHs, flame retardants, and industrial solvents at levels that are harmful to fetal development. We will focus on reviewing what evidence exists to support our hypothesis that WS increases inflammation and oxidative stress leading to endocrine, hemodynamic, vascular, coagulation, and endothelial dysfunction. The underlying mechanism by which WS leads to these derangements may involve epigenetic changes. Each section will summarize the existing body of literature of WS and draw from air pollution data and other combustion related toxicants whenever WS specific data is lacking. 

#### 1.4.1. Inflammation 

Inflammation of the maternal-fetal unit is strongly associated with adverse birth outcomes such as preterm labor, preterm premature rupture of membranes, and neonatal morbidity [58]. It is a well-defined risk factor for prematurity [59]. Thus, the activation of immune cells from triggers can lead to a disruption of the homeostatic balance, which may play an important role in promoting PTB [60]. Maternal inflammatory responses are modified to establish and maintain a viable pregnancy and elevated inflammatory signals are linked with parturition [61]. Imbalances in anti- and pro- inflammatory mediators are associated with common pregnancy complications such as PTB, preeclampsia, and fetal growth restriction. Elevated maternal peripheral blood plasma and/or amniotic fluid levels of pro-inflammatory mediators, such as interleukin (IL)-6 or C-reactive protein (CRP), in mid-pregnancy are associated with spontaneous preterm birth [62,63]. The anti-inflammatory cytokine IL-10 plays a role in maintaining gestation by limiting production of IL-6, tumor necrosis factor (TNF)-α, and other pro-inflammatory cytokines [64]. 

The evidence for WS to induce inflammation specifically during human pregnancy is still growing. WS exposure is significantly associated with inflammatory biomarkers, namely CRP, IL-8, IL-6, peripheral blood segmented-neutrophils, band cells, monocyte chemotactic protein-1, and soluble intercellular adhesion molecule-1, in healthy firefighters actively combating wildfires [65,66,67]. In a recent toxicological study using a first trimester trophoblast cell line, exposure to wood smoke particulates was demonstrated to increase IL-6 secretion in association with decreased secretion of human chorionic gonadotropin (a key placental hormone), increased cytotoxicity, and disrupted mitochondrial structural integrity [51]. A pilot study by our group suggests a dose-dependent increase in fetal Hofbauer cell intravillous infiltration with increasing WS exposure assessed by average daily Air Quality Index measurements throughout gestation, suggesting a placental inflammatory response [68].

Furthermore, WS is a unique contributor ambient air pollution. Several studies have demonstrated an association between a variety of air pollutants and adverse pregnancy outcomes via activation of inflammatory pathways. PM exposure provides the largest body of evidence supporting a systemic inflammatory response to date [56]. For example, PM_10_ pollution was positively associated with serum levels of IL-6, IP-10, macrophage inflammatory protein-1β and eotaxin [69]. PM exposure is also linked to the release of cytokines IL-6 and IL-8 in cell culture studies of human bronchial epithelial cells and alveolar macrophages [70]. Diesel exhaust, which contains PM and PAHs that are found in WS, enhances abundance of blood levels of inflammatory mediators such as IL-1, IL-6, IL-10, IL-12, TNF-α as well as total IgE production by in both animal and human studies [71]. PAH exposure additionally has been found to hyperstimulate maternal immune cells in mouse models [72].

WS human and animal data is starting to emerge. Woodsmoke occupational exposure among firefighters did not have changes in levels of IL-1β, CRP, serum amyloid A, inter-cellular adhesion molecule-1 or vascular cell adhesion molecule-1, but there was an increase in IL-8 [73]. Pregnant rhesus monkeys that were exposed to WS from the 2018 Camp Fire in California during the first trimester showed higher levels of inflammation, blunted cortisol, and elevated C-reactive protein levels [74]. Their exposed offspring also exhibited behavioral changes including more passivity and impaired memory compared to animals that conceived after smoke was no longer present [74]. Exposure in this cohort was defined temporally in relation to when the wildfire event started. Daily PM_2.5_ data from the US Environmental Protection Agency (EPA) for the exposed group was significantly higher due to the 2018 Camp Fire, which was a major wildfire event that destroyed nearly 19,000 structures and contained high levels of phthalates, lead, and zinc due to combustion of houses, cars, and other objects containing plastics [75]. Nasal epithelium samples from WS-exposed macaques were found to have differential expression of 172 genes, which were enriched for pathways involved in leukocyte extravasation signaling, chemokine receptor signaling in macrophages, and migration inhibitory factor regulation of innate immunity [76]. Additionally, decreased pulmonary macrophages were noted in a mouse model 1 h after instillation of wildfire coarse PM_10_ and PM_2.5_ into mouse lung as compared to controls [77]. 

Based on what is known from other related exposures described above, WS likely also impacts the maternal-fetal inflammatory response. It is possible that the extreme exposures during acute wildfire events may lead to increased perturbation of inflammatory mediators that could lead to adverse pregnancy outcomes. Placental inflammation may be the final common pathway for adverse outcomes after wildfire PM_2.5_ exposure, but more comprehensive studies with precise exposure assessments are needed.

#### 1.4.2. Oxidative Stress 

Oxidative stress (OS) is characterized by an imbalance of pro-oxidants like reactive oxygen species (ROS), nitrogen molecules and antioxidant defenses which alter the ability of cells and tissues to detoxify these reactive products [78]. Excessive ROS production can lead to damage at the cellular level due to overpowering the antioxidant defense system [79]. Adverse pregnancy outcomes such as preeclampsia, fetal growth restriction, recurrent pregnancy loss, and spontaneous abortion can develop in response to OS [79]. 

Specific studies on WS exposure have had mixed results regarding OS markers. Among a small group of firefighters pre- and post- work shifts, there were not any significant changes in urinary malondialdehyde or 8-oxo-7, 8-dihydro-2′-deoxyguansoine or any association with biomarkers of oxidative stress measured [80]. Both studies focused on occupational exposure to WS among non-pregnant adult firefighters. More studies are needed to understand the specific impact of WS and systemic OS on the placenta and fetus during pregnancy.

Many investigations have linked air pollutants from various combustion processes to increased oxidative stress in human, animal, and in vitro studies. In animal studies, PM generated from diesel exhaust induces OS by: (1) directly generating ROS due to direct contact with the surface of particles; (2) elevating ROS by exposure to heavy metals or organic contaminants; and (3) altering function of mitochondria, NADPH-oxidase enzymes, and increasing oxidization of DNA and protein/lipid peroxidation [81]. Increased isoprostanes, a marker of oxidative stress, in bronchoalveolar lavage fluid were noted in a mouse model 30 min after instillation of wildfire coarse PM_10_ and PM_2.5_ into mouse lung from control values of 28.1 ± 3.2 pg/mL to 83.9 ± 12.2 pg/mL [77].

Even moderate exposure to PM can induce oxidative lymphocyte DNA damage among non-pregnant subjects [82]. DNA adducts have been shown to be increased in the human placenta, maternal and infant cord blood among pregnancies exposed to air pollution [83,84,85]. DNA adducts are a result from carcinogenic exposures and serve as biomarker of exposure to toxins [86]. PAHs, which are formed during any type of combustion such as coal-fired plants, car exhaust, volcanic eruptions, or forest fires, have been shown to alter biomarkers of OS in plasma and urine from women in their third trimester [87]. Metals such as nickel that are also found in PM have been theorized to inhibit RNA repair enzymes at low, noncytotoxic levels [88]. The chemical 2,6 xylidine, present in cigarette smoke, was found to alter biomarkers IL-6, soluble gp130, and brain-derived neurotrophic factor suggesting an increase in OS in human term placental explant cultures [89].

In a study of 199 healthy pregnant women in Italy, PM_10_ exposure was associated with increased maternal mitochondria DNA copy numbers (mtDNAcn) [90]. These findings are important since cells exposed to OS synthesize more mtDNA to compensate for the damage caused by OS. Among 330 mother-newborn dyads, there was a 35% increase of the level of nitrotyrosinated proteins, a biomarker of OS formed due to the nitration of protein-bound and free tyrosine residues by reactive peroxynitrite molecules, for each interquartile-range increment in PM_2.5_ exposure during pregnancy [91]. This specific biomarker has recently been shown to be a premise of neurodegenerative diseases that is present before the onset of symptoms, since the nitration of proteins results in damage to neuronal cytoskeletons [92]. In a cohort of 160 pregnant participants in California, a recent study demonstrated in a pathway enrichment analysis that the prostaglandin pathway and fatty acid, phospholipid, lineoleate, and eicosanoid metabolism were significantly altered in the maternal serum metabolome with exposure to traffic-related air pollution during pregnancy [93]. These pathways alterations suggest associations between air pollution exposure and OS and inflammation [93]. The evidence for OS resulting from air pollutants is robust; however, WS specific exposure data has yet to be evaluated in the pregnant population highlighting the need for further research.

#### 1.4.3. Endocrine Dysfunction

Throughout pregnancy, the placenta plays a critical role in development, growth, and ultimately, the survival of the fetus [94]. The placenta secretes many proteins, hormones, and bioactive lipids involved in the regulation of transfer and fetal development across gestation. The transfer of these nutrients are all critical for the fetus in the short term and the long term and though to contribute to the early origins of disease later in life as described by the Barker hypothesis [95]. 

Although WS has not yet been studied specifically, air pollution and PM exposures appear to lead to changes in placental proteins, transporters, hormones, and bioactive lipids involved in crucial fetal and placental pathways. In non-pregnant individuals, short term exposure to ambient PM led to significant changes in oxylipins derived from polyunsaturated fatty acids from LOX, CYP and COX pathways [96]. In utero exposures to PM specifically during the second trimester was associated with differences in the cord blood levels of metabolites derived from the lipoxygenase pathways among 222 mother-newborn pairs from the ENVIR*ON*AGE cohort [97]. These findings are consistent with animal models which demonstrated that diesel exhaust exposure led to elevated concentrations of oxylipins (5-HETE, 12-HETE, and 13-HODE) from the 5-LOX and 12/15-LOX pathways [98]. In a pregnant mouse model of PM exposure, the proliferative capacity of placenta cells was disrupted and mRNA expression of amino acid, long-chain polyunsaturated fatty acid, glucose, and folate transporters [99], suggesting PM may impact endocrine signaling and compromise metabolism and transport.

Thyroid hormones are crucial for fetal growth and development and the ENVIR*ON*AGE birth cohort demonstrated that PM_2.5_ exposure is inversely related to cord blood thyroid stimulating hormone and free thyroxine/free triiodothyronine ratio [100]. These alterations in thyroid hormone levels led to a 56 g decrease in mean birth weight in this birth cohort of over 400 mother-infant pairs. Thyroid function alterations have also been reported among pregnant cohort of 659 women in Puerto Rico exposed to PAHs in addition to alterations in estriol, progesterone, and testosterone [101]. 

While tobacco smoke is different than WS in its chemical composition, there are similarities given it is a product of combustion—many of the chemicals are found in both smoke sources. Human placental tissue from smokers was studied and noted to have markedly decreased epidermal growth factor-stimulated kinase activity, an important component of placental growth and differentiation, when compared to non-smoker tissue [102]. Benzo[a]pyrene, a PAH that is ubiquitous in combustion products, was also found to inhibit epidermal growth factor in early gestation placentas suggesting a role in the regulation of placental growth among pregnant patients exposed to smoke [103]. It has been suggested that the carcinogenic fraction of PAHs (c-PAHs), which are usually bound to fine particles from air pollution, may directly modulate the proliferation of the placental trophoblast due to their reaction with placental growth factor receptors [104]. Dejmeck and his colleagues’ work noted that for every 10 ng increase of c-PAHs in the first gestational month, the adjusted odds ratio was 1.22 (95% CI 1.07–1.39) for intrauterine growth restriction [104]. Importantly, we emphasize that WS in pregnancy has not yet been studied specifically with regard to endocrine dysfunction. However, the evidence for particulate matter exposure and chemicals from other smoke sources appear to influence placental proteins, transporters, and hormones in harmful ways. Thus, we posit that WS is similarly implicated in adverse pregnancy outcomes, especially related to growth processes based on available data. 

#### 1.4.4. Hemodynamic and Vascular Function

Hemodynamic and vascular functional changes may represent additional pathways which WS exposure contributes to harm. Altered vascular and endothelial function has been shown to be associated with preterm birth, hypertensive disorders of pregnancy, preeclampsia, and fetal growth [105]. In pregnancy, a drastic redistribution of blood flow is needed to sustain healthy fetal growth. Accordingly, changes in vascular and angiogenic factors can lead to abnormal placental development and pregnancy complications [105].

WS exposure during pregnancy has been linked with hypertensive diseases in pregnancy [29]. A 1 μg/m^3^ increase in PM_2.5_ exposure to WS during the first and second trimester or over the full gestation was positively associated with gestational hypertension [29]. Similarly, another study found that for each 0.52 μg/m^3^ increase in delta-C concentration, an environmental marker for wood combustion, during the 7th and 8th gestational month was associated with an elevated risk of hypertensive disorders of pregnancy, including risk for early onset preeclampsia [39]. 

Studies of non-pregnant individuals on the inhalation in urban settings of ambient particles and ozone for 2 h led to arterial vasoconstriction in healthy adults [106]. Ozone exposure during implantation appears to induce fetal growth restriction due to decreased uterine blood flow and is used to generate models of fetal growth restriction in pregnant Long-Evans rats [107]. Acute systemic inflammation and oxidative stress following PM exposure, as described above, are likely responsible for triggering endothelial dysfunction leading to vasoconstriction [56,82,108].

PM and products of combustion from wildfire specific smoke likely interact with maternal blood pressure given vascular inflammation and oxidative stress are considered to lead to hypertension [109]. A potential mechanism whereby pollutant components can increase blood pressure is superoxide-mediated inhibition of the actions of nitrous oxide in inducing vasodilatation [8]. A systematic review by Bekkar et al. (2020) investigated prenatal exposure to PM_2.5_, ozone, and heat. They found a significant association with air pollution, heat exposure, and adverse birth outcomes such as preterm birth and low birth weight in a total of 57 studies reviewed. The underlying mechanism is hypothesized to be reduced uterine blood flow that alters placental-fetal exchange of nutrients [110]. In a rabbit model, diesel engine exhaust exposure in pregnant dams was noted to reduce placental efficiency, decrease placental blood flow, and increase umbilical artery resistance [111].

Tobacco smoke is known to rapidly cause vasoconstriction, increase plasma endothelin levels, and trigger endothelial dysfunction [112,113,114]. One major product of combustion, and tobacco smoking, is carbon monoxide (CO). Investigations of umbilical cord arterial blood from pregnant cigarette smokers had threefold higher CO concentration compared to non-smokers [115]. CO is capable of significant decreases in placental perfusion pressure, indicating that CO found in the serum of smokers is capable of hemodynamic control within the placenta [115].

WS exposure likely disrupts vascular and endothelial regulation in the placenta and therefore contributes to significant morbidity via hypertensive disorders of pregnancy such as preeclampsia, though further study is needed given the limited studies available for review.

#### 1.4.5. Coagulation

Coagulation derangements may represent a separate mechanism by which WS related PM contributes to toxicity in pregnancy. Pregnancy is a hypercoagulable state wherein there is a normal significant decrease in physiologic anticoagulants [116]. Imbalances in the hemostatic system can lead to uteroplacental thrombosis and thus reduced placental perfusion [116]. Uteroplacental under perfusion is linked to adverse pregnancy outcomes such as recurrent pregnancy loss, preeclampsia, and fetal growth restriction [116,117].

WS exposure from the 2018 Camp Fire in felines demonstrated increased risks in clot formation with the suggestion that platelet priming and activation may contribute to a global hypercoagulable state and thrombosis [118,119].

It is well established that PM exposure in adults results in increased whole blood viscosity, coagulability, proteins related to the clotting cascade, and coagulopathic factors such as factor VII-IX, fibrin D-dimer, and von Willebrand factor [120,121,122,123,124,125]. A cross-sectional study conducted in London found a significant association during the warm season between PM_10_ and plasma fibrinogen [120]. Alterations in the levels of hemoglobin, platelets, white blood cells due to exposure to PM have also been reported in healthy, non-smoking young men [126]. Among pregnant cigarette smokers, there is an imbalance between fibrinolysis and coagulation, as assessed by the D-dimer/Thrombin-antithrombin III ratio, which suggests that tobacco smoke exposure leads to a hypercoagulable state [127]. In mice exposed to concentrated PM pollution episodes in the San Joaquin Valley of California, platelets were significantly elevated in number and a 54% increase in fibrinogen binding was seen, suggesting platelet priming (a coagulopathic process) as a result of severe PM exposure [128].

These studies suggest that components of WS may impact coagulation pathways that contribute to disease processes with pregnancy, adding an additional layer of complexity during a time when the balance between procoagulants and anticoagulants is already skewed toward a hypercoagulable state. The relationship between WS exposures and changes in coagulation have not been studied in context of pregnancy, however. This further highlights the necessity of future research in this area.

#### 1.4.6. Epigenetic Alterations

Alterations in DNA methylation may provide a unifying mechanism by which exposure to air pollution and WS may lead to downstream effects on genes and protein synthesis. Epigenetic regulation of the placenta evolves throughout pregnancy starting from preimplantation onwards [129]. Alterations in placental epigenetics have been associated with pregnancy complications such as fetal growth restriction, small for gestational age, pre-eclampsia, and gestational trophoblastic disease [129,130,131].

DNAm refers to the methylation of a DNA molecule (DNAm), often at CpG (cytosine precedes guanine) sites. Differentially methylated DNA molecules have changes in gene expression and have the potential to reflect past exposures with long-term effects [76]. Hypomethylation is associated with genomic instability via chromatic structure modeling and increased oncogenic activation, which may underlie the mechanistic processes that lead to cellular dysfunction [132,133].

Adults exposed to wildfires have been shown to have decreased methylation at the PDL2 gene, a programmed death ligand, which plays an important role in downregulating T cell responses important to immunological function and associated with airway inflammation [134]. In children exposed to WS there is an increase in Foxp3 methylation in the promoter region of DNA from blood samples and a reduction in the Th1 pro-inflammatory T cells [135]. Additionally, there was a trend toward poorer health with increases in wheezing and asthma exacerbations [135]. Adult and childhood human studies are consistent with the latest animal models. In a recent study, there were notable differences in methylation and gene expression in nasal epithelial samples among macaques that were exposed to during the 2008 California wildfire season early in life compared to macaques that were not exposed [76]. The WS associated differentially methylated regions were related to genes for synaptogenesis signaling, protein kinase A signaling, and a variety of immune processes [76].

Exposure to air pollution has been demonstrated to affect the methylation of genes that code for inflammatory mediators, including interferons and interleukins [136,137,138,139]. Maternal and cord blood long interspersed nuclear element-1 (LINE-1) methylation levels were negatively correlated with prenatal exposure to PM and risk for preterm birth [140]. LINE-1 downregulation can induce an inflammatory response and thus is hypothesized to be important for initiating PTB cascade [140]. Pregnancy studies investigating the methylation of the aryl hydrocarbon receptor repressor (AHRR) gene, which mediates metabolism of smoke toxins, indicate that increased levels of PM_2.5_ in non-smokers were associated with decreased levels of AHRR methylation, thereby illustrating an inverse association between DNAm and PM_2.5_ levels [141]. In a study of 20 mother-newborn dyads, statistically significant differences were noted for 31 CpG sites associated with 25 genes among mothers that smoke tobacco in umbilical cord blood [142]. The most significant site located within the adrenomedullin gene (chromosome 11p15.4) which is important in the regulation of hormones and cell growth [142]. 

Placental studies have shown that exposure to tobacco during pregnancy is associated with lower birth weight and lower relative mitochondrial DNA which can be used as a marker of mitochondrial damage and dysfunction [143]. CpG-specific methylation levels of cytochrome P450 family 1 subfamily A member 1 (CYP1A1) have been shown to decreased in placental tissue but increased in cord blood in response to smoke exposure [144]. CYP1A1 is implicated in the metabolism of PAHs and aids in detoxification processes [145]. Air pollution PM_10_ and NO_2_ exposure has also been demonstrated to lead to locus-specific DNA methylation alterations in the placenta [146]. One of the loci discovered was near the gene for adenosine A2b receptor, which is associated with hyperglycemia and OS in conditions such as gestational diabetes mellitus [147,148]. More targeted investigations in pregnancy are needed to understand which genes and proteins are most affected by WS; however, the current limited evidence supports harmful DNAm changes in a variety of pathways mostly related to inflammation.

#### 1.4.7. Telomere Length

As of this review, there is not yet wildfire specific evidence for telomere shortening. However, there is burgeoning evidence for prenatal exposure to air pollution and tobacco smoke. Telomeres are repetitive DNA sequences that cap the ends of chromosomes, critical in preventing the loss of coding regions during chromosomal replication. They have been proposed to play a role in fetal programming and health outcomes [149]. Telomere length is linked to biological age and is impacted by oxidative stress [90]. In a cohort of 199 healthy pregnant women in Italy, it was demonstrated that PM_10_ exposure in the first trimester is associated with reduced telomere length and reduced birth weight in offspring [90]. In term cord blood studies, tobacco smoke exposure led to shortened fetal telomere length from genomic DNA isolated from leukocytes [150]. In placental tissue and fetal lung tissue, there have been reports of shortened telomere length in pregnancies exposed to tobacco smoke [151]. Taken together, these studies suggest that tobacco smoke and particulate matter exposure can lead to shortened telomere length, affecting early intrauterine programming, fetal neural development, and changes to gene expression while accelerating the aging process. Notably, we could not identify any wildfire specific investigations on telomere length during pregnancy. However, we propose that WS is likely to decrease telomere length in a similar fashion to other air pollutants.

## 2. Review of the Current Human, Animal and Placental Studies

As described above, we summarized the most compelling available human, animal, and placental studies with WS and other combustion products such as tobacco, diesel, and wood smoke in Table 1, Table 2 and Table 3. Notably, these tables are not exhaustive lists; however, they are representative of the current literature available and provide an overview of the most salient investigations. In the next section of this review, we will summarize the current recommendations and interventions that are available to pregnant individuals seeking to avoid the harms of WS.

### 2.1. Exposure Reduction Strategies and Implications for Research

Public health authorities such as the US Centers for Disease Control and Prevention (CDC), EPA, and Statewide Departments of Public Health have issued recommendations to reduce WS exposure. Among the most important strategies: (1) staying indoors with a high-efficiency air filter, (2) seeking shelter with a high-efficiency air filter, (3) using a N95 or P100 respirator, and (4) reducing outdoors exposures and avoiding strenuous activities to reduce smoke inhalation. The other strategies are summarized in Table 4 below [3]. These strategies are especially relevant to the most vulnerable populations: children, people with asthma, chronic obstructive pulmonary disease, heart disease, or pregnancy [158].

#### 2.1.1. Behavioral Modifications, Air Filters, and Masks

Behavior modifications have the potential to reduce exposure, but the health impacts of behavior modifications have not yet been demonstrated [159]. The primary recommendation for all persons to stay indoors and avoid strenuous activity is based on epidemiologic evidence of PM exposure from wildfire events and the resultant health outcomes (e.g., mortality, respiratory morbidity, asthma, etc.) [160]. If evacuation is necessary, finding a shelter that has high quality air filtration is key. The use of high efficiency particulate air (HEPA) filters and well-fitting respirators (N95 or P100) in the general population have been shown to reduce PM exposure, translating into improvements in cardiovascular and respiratory health indicators such as blood pressure, endothelial function, and systemic inflammation [159,161,162]. Furthermore, due to particulate deposition of toxins and carcinogens, it is advised to not consume any food or beverages that have been exposed to ash and burn debris as well as to avoid creating further sources of pollution such as from stoves, fireplaces, or other combustibles.

Despite these recommendations, there are significant structural barriers to their widespread implementation. For example, having a low socioeconomic status could increase exposure to WS due to higher rates of outdoor work, decreased accessibility to quality shelter/housing, decreased accessibility to respirator fit-testing, inability to purchase air filters or respirators, barriers to information and healthcare, and a limited ability to evacuate [163,164]. Thus, these recommendations should be reviewed while carefully considering health equity and potential confounders to health outcomes.

#### 2.1.2. Personal Air Quality Monitors

Another potential strategy to reduce smoke exposure is to leverage personal monitoring devices and modifying individual behavior based on real-time exposures. Wearable device air quality monitoring provides highly accurate air pollution exposure measurements, since people are mobile throughout the day; however, this method can be tiring and costly, resulting in high rates of noncompliance [165]. Most people, including specifically pregnant women, spend the majority of their time indoors and their indoor pollution estimates were found to be similar to wearable monitor measurements, making indoor air quality monitoring a practical and effective alternative to wearable device monitoring for both research purposes and individual knowledge of exposure [165,166,167]. Increasing weeks of pregnancy is a significant predictor for increased time spent at home (1 h/day increase for each trimester of pregnancy), after adjusting for income (2.6 more h/day at home in lowest income group), work status (3.5 more h/day at home for nonworkers) and other children in the family (1.5 more h/day at home with other children) [166]. A cohort study showed that 71% of 163 children wearing a monitor reported that their activity was representative of normal activity in non-wildfire situations [158]. This may mean that wearable monitoring and subsequent behavior modifications are likely insignificant in reducing air pollution exposure in non-wildfire situations; but to the best of our knowledge, no personal monitoring studies (either community or wearable) have been conducted in wildfire settings. Thus, it remains to be determined whether personal monitoring would help modify behavior and reduce WS exposure.

It is currently not known which interventions and prevention methods are being utilized by pregnant people to reduce WS exposure, nor the benefits and costs of each intervention in this population. A randomized controlled trial in pregnant people found that HEPA filter air cleaner use in a high air pollution setting (Ulaanbaatar, Mongolia) during pregnancy was associated with greater birth weight among babies born at term [168]. A systematic review found that limited duration N95 use is a relatively accessible and benign intervention if fit-testing is available—it was unlikely to impart risk to a pregnant person or the fetus and repeat fit testing in pregnancy is unnecessary in the absence of excess weight gain [169]. There were also no significant differences in measured physiological and subjective responses in pregnant versus nonpregnant people wearing an N95 respirator for 1 h during both exercise and sedentary activities [170]. To the best of our knowledge, the impact of N95 respirators on WS exposure’s effects on maternal and fetal outcomes has not been studied. Overall, assessment of these interventions to reduce WS exposure in pregnant populations and children are much needed.

#### 2.1.3. Implications for Research

The interventions to reduce exposure during real-time wildfire events are individually implemented, and thus the effects are difficult to assess in retrospective, population-based studies. Individual and group variations in the use of these strategies may explain the conflicting data reported by epidemiological studies thus far, as outdoor air quality measurements are unlikely to capture the nuances of individual-level exposures that could have a dramatic impact on pregnancy outcomes.

Vulnerable populations are at particular risk for WS exposure and the associated adverse pregnancy outcomes, amplifying existing reproductive health disparities. The associations between air pollution and preterm birth are strongest in neighborhoods with lower socioeconomic status. People of color and low socioeconomic status may have greater exposures and worse responses to wildfire PM_2.5_ due to higher rates of outdoor work, living in poor-quality housing, barriers to information access, limited evacuation abilities, and structural barriers to healthcare access [163]. This double jeopardy of exposure to pollutants and poverty is hypothesized to contribute to disparities in adverse pregnancy outcomes. Unfortunately, robust data have not yet been published. Accurate measurement of individual-level exposures is needed to clearly define exposure risks, as indoor measurements available through home air quality monitors are most likely providing data from affluent households that can afford home monitors, thus not capturing higher risk population exposures.

The impact of WS on birth weight and when the most critical periods of exposure during pregnancy are yet to be fully established. Much of the data on air pollution and pregnancy outcomes are derived from measured and modeled outdoor air quality data from stationary monitors and satellite imaging. Importantly, these data do not account for behavioral, environmental, or neighborhood factors that can influence individual exposures to smoke. Living and working conditions, use of air purifiers or masks, or temporary relocation may dramatically impact individual exposures in wildfire events beyond that suggested by outdoor air quality reports. In some areas, individual indoor monitors (e.g., PurpleAir) can provide additional data, but these are limited to households of higher sociodemographic status [171]. Thus, to understand the biologic effects specific to WS exposure, prospective cohorts with individual-level data are needed, especially in higher risk communities.

## 3. Discussion

This review examined the limited body of evidence for WS exposure and potential biological mechanisms by which it may impact the placenta and adverse pregnancy outcomes. It is important to note that this is not a systematic review given wildfire specific biologic models and studies are just beginning to emerge. Given the field is in its nascency, we hope this review highlights that further study is much needed. A summary of the concepts discussed in this review is represented in Figure 2, below.

The strengths of this review include a focus on multiple disciplines and methodologies ranging from epidemiologic studies to laboratory-based studies with various biochemical and molecular techniques. We also included studies that were WS specific as well as other exposures that we believe are similar, such as air pollution, particulate matter, and other chemicals related to combustion. The limitation with this approach is that the assessment of exposure to WS or other pollutants was significantly variable within each study; some exposures were quantifiable while others were limited to time and space considerations. The heterogeneity of the populations studied represents a challenge in the interpretation of the existing body of literature given human pregnancy data and the animal models available are limited. Importantly, we underscore the need to better understand in granular detail how WS affects the placenta and pregnancy.

## 4. Conclusions 

We provide a review of the current limited literature surrounding placental biology and biological mechanisms from which adverse pregnancy outcomes may arise from due to exposure to WS. Since WS specific data is still being generated, this review also incorporated what is known about the placental impacts from general air pollution and other combustion products in animal and human models. We also captured current recommendations of strategies to avoid WS exposure in pregnancy. From what the current body of evidence demonstrates, WS potentially affects a variety of critical processes including: inflammation, oxidative stress, endocrine disruption, DNA damage, telomere shortening, epigenetic changes, as well as vascular, endothelial, and coagulation dysregulation of the maternal-fetal unit. We note that more individualized exposures assessments are needed to quantify WS exposure in pregnant individuals. Future investigations, both in vitro and in vivo will be critical to closely examine and dissect the biological mechanisms that underpin adverse birth outcomes caused by WS exposure in pregnancy. Understanding these crucial mechanisms will ultimately tailor preventative strategies, guide future therapeutics, and help bridge the gap of worsening maternal and fetal morbidity in the most vulnerable populations.

## Figures and Tables

**Figure 1 ijerph-19-13727-f001:**
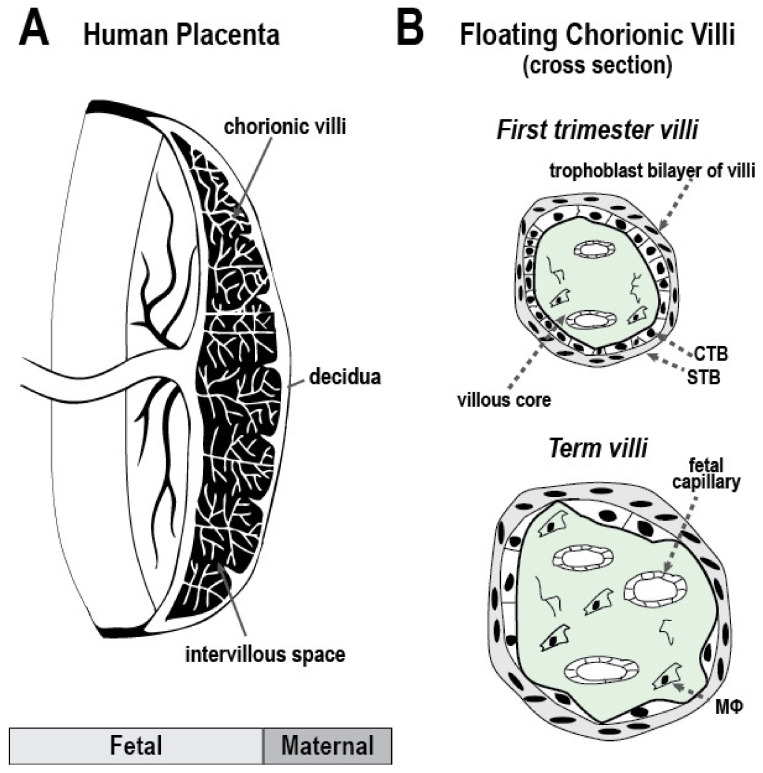
Human placenta and chorionic villi. The placenta (**A**) is the site of xenobiotic transfer from the mother to the embryo/fetus—an important barrier for environmental chemicals. Chorionic villi (**B**) are covered by an outer layer of fused, multinucleated syncytiotrophoblasts (STB) and an inner layer of villous cytotrophoblasts (CTB). The villous core consists of the basal lamina, mesenchymal cells, residing Hofbauer cells (MΦ), and fetal capillaries. The architecture and function of the placenta evolves over gestation to support the fetus. In the first and second trimesters, a trophoblast bilayer lines the villus, and invading CTBs differentiate to anchor and establish placental blood flow from the uterus. By the third trimester, nutrient exchange is primary, as well as the support and protection of the fetus. At term, chorionic villi contain more capillaries and immune cells, and fewer invasive CTBs. The legend on the bottom left of (**A**) denotes the fetal-facing amnion (light grey) and maternal-facing chorion (dark grey).

**Figure 2 ijerph-19-13727-f002:**
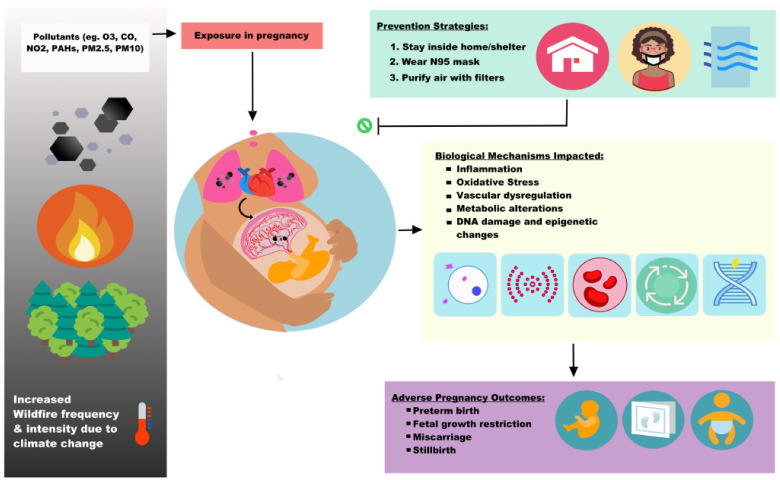
Wildfire smoke exposures and pregnancy implications. As climate change elevates global temperatures, wildfires are becoming more frequent, intense, and destructive. Wildfire smoke (WS) is a complex mixture of particulate matter (PM), gases such as ozone (O_3_), carbon monoxide (CO), nitrogen dioxide (NO_2_), polycyclic aromatic hydrocarbons (PAHs), heavy metals, elemental and organic carbon, in addition to volatile and semi-volatile organic compounds. Pregnant persons are especially susceptible to the adverse health effects of WS pollution due to the increased respiration and cardiovascular output in pregnancy. WS exposures in pregnancy can impact multiple biological mechanisms and contribute to inflammation, oxidative stress, vascular and endothelial dysfunction, metabolic alterations, and DNA damage as well as epigenetic changes. These alterations can lead to adverse pregnancy outcomes such as preterm birth (PTB), fetal growth restriction, and pregnancy loss. This highlights the importance of prevention strategies such as staying indoors, wearing a high-quality mask, and air filtration.

**Table 2 ijerph-19-13727-t002:** Summary of animal studies included in our review. For each study, we provide the citation, type of exposure, time frame, location, animal model, and sample size. We also provide a description of the exposure. Finally, we indicated the primary outcome observed. Citations are listed in alphabetical order.

Citation	Type of Exposure	Time	Location	Animal Model	Sample Source(s)	Sample Size (*n*)	Description of Exposure	Primary Outcome(s)
Black (2017) [26]	Wildfire smoke	2008	California, USA	Adult female rhesus macaque (Macaca mulatta) exposed in infancy to wildfire season in 2008	Blood	50	Ozone and PM_2.5_ concentrations from air monitoring stations	PBMC in vitro challenge testing, cytokine protein assay 6 h after TLR ligand addition, IL-6 and IL-8 levels, Pulmonary function measures
Brown (2022) [76]	Wildfire smoke	2008–2009	California, USA	Adult female rhesus macaque (*Macaca mulatta*) exposed in their first three months of life to wildfire season in 2008	Nasal epithelium, peripheral blood	22	Wildfire smoke PM_2.5_, ozone exposure during early life	Whole genome bisulfite sequencing to identify differentially methylated regions from nasal epithelium, RNA-sequencing on a subset of samples
Capitano (2022) [74]	Wildfire smoke	2018	California, USA	Infant female rhesus macaque (*Macaca mulatta*)	---	56	Mean daily PM_2.5_	BioBehavioral Assessment, CRP levels via high sensitivity assay, cortisol concentration via I125 radioimmunoassay
Detmar (2008) [72]	Polycyclic aromatic hydrocarbons	2008	Toronto, Canada	C57BI/6 female mice	Placenta, fetus	4	Subcutaneous injections of PAH over a 9-week period	Fetal growth, placental cell death rates, expression of antiapoptotic Xiap, proapoptotic Bax, levels of cleaved poly(ADP-ribose) polymerase-, and active caspase-3
Hong (2013) [152]	Particulate matter	2012	Fujian, China	Mouse	Blood, spleen, thymus	40	Instillation of airborne PM solution into mouse lung	IL-4 and IFN-*γ* levels in plasma and spleen, splenic lymphocyte proliferation, GATA-3 and T-bet mRNA in spleen tested, histopathology of spleen and thymus.
Lee (2015) [153]	Environmental tobacco smoke	2015	California and Montana, USA	Mouse	Bronchial alveolar lavage fluid	4	Tobacco smoke (1.0 mg/m^3^) for 6 h/day	Global DNA methylation, cytokine measurements
Valentino (2016) [111]	Diesel engine exhaust	2016	France	New-Zealand white female rabbits (INRA1077 line)	Maternal lung, maternal and fetal plasma, placenta, fetus	28	Inhalation of diesel exhaust from 3rd to 27th day post-conception (20 days over 31 day gestation)	Ultrasound with biometry and Doppler monitoring, birth weight, TEM of lung, vascular and placental tissue to identify NPs
Miller (2020) [107]	Ozone	2020	North Carolina, USA	Long-Evans rat	Placenta, fetus	8	Gaseous ozone for 4 h in the mornings of gestation days 5 and 6 (during implantation)	DNA and RNA expression from placenta, hepatic gene expression, mitochondrial respiration, metabolic assessment
Zhu (2021) [99]	PM_2.5_	2021	Taiguan, China	C57BL/6 mice	Placenta, fetus	55	Oropharyngeal aspiration of PM_2.5_ every other day starting on embryonic day 0.5	Expression of proliferating cell nuclear antigen, mRNA of amino acids, long-chain polyunsaturated fatty acid, glucose, glycogen, triglycerides, and folate transporters

Abbreviations: PAH = polycyclic aromatic hydrocarbons; PBMC = peripheral blood mononuclear cell; TEM = transmission electron microscopy.

**Table 3 ijerph-19-13727-t003:** Summary of human placental studies included in our review. For each paper, we provide the citation, time frame, location, study design, study population, and sample size. We also provide a description of the exposure. Finally, we indicated the primary and secondary outcomes observed, if applicable. Citations are listed in alphabetical order.

Citation	Type of Exposure	Time	Location	Study Population	Sample Source(s)	Sample Size (*n*)	Description of Exposure	Primary Outcome(s)
Abraham (2018) [146]	Air pollution	2003–2006	France	Singleton pregnancies enrolled before 24 weeks of gestation	Placenta samples collected at delivery	688	NO_2_ and PM_10_ hourly concentrations modelled using maternal home address, mean daily temperature and humidity from nearest stationary monitors	Genome-wide DNA placental methylation levels
Adebambo (2018) [154]	Cadmium treatment	2018	Longjiang River, China	JEG-3 choriocarcinoma cell line	Placental trophoblast cells	6	Environmental water samples from a cadmium spill site in China	Reactive oxygen species (ROS), expression of metallothionein (MT) isoforms, HIF1α, and TGFβ associated genes and proteins
Arita (2022) [89]	Dimethylaniline (DMA)	2022	New York, New Jersey, USA	Placental explants	Placental explant cultures from term placentas collected from elective cesarean sections	12	DMA instilled into placental explant well plates to final concentrations of 0–50 μM	IL-1β, TNF-α, IL-6, sgp130, IL-10, BDNF, HO-1, 8-IsoP, P4, T, E2 quantification using immunoassay reagents
Basilio (2021) [68]	Wildfire smoke	2018–2019	California	Mid gestation placenta	First and second trimester placenta from elective terminations of pregnancy	12	Average daily AQI levels during gestation	Fetal Hofbauer cells (CD68+)
Bové (2019) [49]	Air pollution	2012–2016	Belgium	ENVIR*ON*AGE birth cohort, singleton births recruited on arrival for delivery	Term and preterm placenta collected within 10 min after birth	10	Ambient exposure to BC determined using maternal residential address using spatial and temporal integration from satellite images and pollution data from fixed monitoring stations	Detection of BC particles, BC load
Bainbridge (2006) [115]	Carbon monoxide	2006	Canada	Placental explants	Placental explant cultures from term placentas collected from elective cesarean sections	13	Carbon monoxide infused culture medium	Apoptotic Index using TUNEL assay, Immunohistochemical staining for p85 Fragment of PARP, morphology of villous tissue, villous tissue integrity
Erlandsson (2020) [51]	Wood smoke	2020	Sweden	First trimester trophoblast cell line HTR-8	HTR-8 was derived by transfecting cells from chorionic villi explants from placentas of 6–12 week gestation	6-well plates	Smoke burned on wood stove from logs of four different species (silver birch, quaking aspen, Norway spruce, and Scots pine) at a nominal burn rate and high burn rate, wood particles collected and extracted then aliquoted into well plates	hCG, progesterone and IL-6; cellular particle localization and uptake by TEM, cytotoxicity assay, PAH analysis, membrane integrity testing
Janssen (2012) [155]	Air pollution	2012–2016	Belgium	ENVIR*ON*AGE birth cohort, singleton births recruited on arrival for delivery	Human, placenta and cord blood	174	PM_10_ incremental exposure	mtDNA content
Janssen (2013) [156]	Air pollution	2012–2016	Belgium	ENVIR*ON*AGE birth cohort, singleton births recruited on arrival for delivery	Human, placenta	240	PM_2.5_ incremental exposure	Global DNA methylation
Janssen (2015) [157]	Air pollution	2012–2016	Belgium	ENVIR*ON*AGE birth cohort, singleton births recruited on arrival for delivery	Human, placenta	381	PM_2.5_ incremental exposure	mtDNA methylation
Saenen (2015) [57]	Air pollution	2012–2016	Belgium	ENVIR*ON*AGE birth cohort, singleton births recruited on arrival for delivery	Human, placenta	90	PM_2.5_ incremental exposure	Gene expression in BDNF and SYN1 pathways
Saenen (2016) [91]	Air pollution	2012–2016	Belgium	ENVIR*ON*AGE birth cohort, singleton births recruited on arrival for delivery	Human, placenta	336	PM_2.5_ incremental exposure	3-nitrotyrosine

Abbreviations: BC = black carbon; TEM = transmission electron microscopy; PM = particulate matter; mtDNA = mitochondrial DNA.

**Table 4 ijerph-19-13727-t004:** Prevention strategies to reduce wildfire smoke exposures for vulnerable populations.

Prevention of WS Exposure
(1) Stay indoors with a high-efficiency air filter *
(2) Seek shelter with a high-efficiency air filter *
(3) Use N95 respirator or P100 respirator
(4) Reduce outdoor exposure
(5) Reduce strenuous activities to reduce inhalation
(6) Evacuate safely and prepare an evacuation kit with food, water, and medications for 7–10 days, first aid supplies and important documents
(7) Do not consume any food, beverages, or medications that have been exposed to burn debris or ash; avoid using wood-burning stoves, fireplaces, gas, propane, or vacuum
(8) Protect pets by keeping them indoors. If you must evacuate without your pets, never tie them up

* High Efficiency Particulate Air rating (HEPA) or Minimum Efficiency Reporting Value rating (MERV) of 17–20.

## Data Availability

Not applicable.

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
