# Peer review of "Wildfire Smoke Exposure during Pregnancy: A Review of Potential Mechanisms of Placental Toxicity, Impact on Obstetric Outcomes, and Strategies to Reduce Exposure"

_ijerph, 2022, doi:10.3390/ijerph192113727_

Round 1
Reviewer 1 Report
· I suggest numbered headers for each section. It is not easy to follow the sequence of sections/headers in the current format.
· In general this is a well written paper, but there are a number of grammatical errors throughout. I suggest a thorough readthrough prior to resubmission/publication.
· See Burke et al. 2021 “The changing risk and burden of wildfire in the United States” for more up to date numbers on wildfire contribution to total PM emissions in the US.
· Define every abbreviation on 1st use – PM defined on second use. Use the abbreviations more consistently – there are a few instances where you switch back and forth between WS and wildfire smoke.
· Clarify this sentence: “Complications of severely term birth are directly correlated with the gestational age at which neonates are born”
· Suggest adding some citations in the 1st paragraph under section: Anatomy of the human maternal-fetal interface and particulate matter deposition
· Figure 1: it is not clear what the “Fetal” and “Maternal” key at the bottom on the figure is referring to.
· Suggest using different language when distinguishing WS and other sources of ambient air pollution. For example, in this sentence “Key components of wildfire smoke are also found in air pollution and our review of the body of literature on air pollution shows an association with exposure to adverse birth outcomes and pregnancy complications via inflammatory pathways.” It is confusing because WS is air pollution. Maybe rephrase air pollution as “other sources of ambient air pollution” in this sentence and throughout.
· The glaring weakness I see in this review is that many of the studies you highlight are not specific to either 1) wildfire exposures or 2) placental/obstetric outcomes. I get that there are limited studies on wildfires and these outcomes of interest and you are trying to fill in the gaps using studies from other exposures/populations. However, as you have presented it in this draft, the focus on wildfires in the title, abstract, and discussion/conclusions is quite misleading. This is not so much a review on wildfire exposures but air pollution exposures in general. In nearly every section, you cite work from other areas of research, state that research specific to wildfires is limited, and postulate that the findings from other studies may also apply to wildfires.
· The section on exposure reduction strategies seems tacked on at the end of this paper and does not add a lot. There are numerous other papers that discuss exposure reduction.
· Overall, this paper has a misleading focus/conclusions on wildfire smoke. As it is, I can’t recommend this draft for publication. However, there is a lot of good/important information in this paper. I think it could be edited and rebranded as an updated review on overall air pollution and physiological pathways related to pregnancy outcomes. This could include a call for more research on wildfires and pregnancy outcomes, which is important due to the different composition of wildfire smoke and other air pollution sources that are clearly stated in this draft already.
Reviewer 2 Report
Keep up the good work.
I encourage the authors to collect similar up-to-date data about the effect of the current wildfire burnings in the state of California as we read and write these lines.
Page 1,
I can’t see any mistakes to mention in this page except the following:
On the top of the page, it said (2021, 18, For Peer Review), should not the year be 2022! Instead of 2021?
UC-Berkeley2 (School of Public Health, University of California Berkeley). Why the authors did not complete the address to (Berkeley, CA 94720-7360)?
Page 2, Last paragraph they wrote (Polycyclic aromatic hydrocarbons (PAHs).
The abbreviation “Polycyclic aromatic hydrocarbons (PAHs)”, as written by the authors, found in many pages needs to be included in the abbreviation section. Needs to be identified on page 2 first. It was spelled out in page 7 after it has been used many times with no definition.
Please define Polycyclic Aromatic Hydrocarbons (PAHs) with capital letters as shown not lower-case letters on this page2 then, reuse them throughout manuscript.
Page 3, the authors wrote (Of the 7 studies on wildfire smoke and birth weight, 6 found an association with decreased birth weight [32, 35, 37, 38, 39,40,41].)
As you can see above these references has some punctuation adjustments. All need to be written with no spaces after the comma. As such [32,35,37,38,39,40,41]. Or, as the journal guide to authors stated.
Also, under Page 3 the authors wrote under Fetal Growth the following: (These wildfire-specific studies are summarized in Table 1. For example, exposure to WS PM2.5 in Colorado…) then, when the reader looks up only Table within the manuscript, it appears on page 12!
Pages 4-7
Again, on page 5, the following references ([50,51, 52,53,54] has extra spaces or no spaces between them, be consistent in punctuation and presenting the references.
A e very good points made in this review that will trigger new research ideas/directions such as when the authors wrote on page 5: (These findings establish that the placental barrier is not impenetrable to particulate matter from air pollution exposures, although the extent of “transmission” to the fetus remains unknown.)
Keep up the good work.
Under inflammation, in page 6, what are these “CCR5” and “MIF” abbreviations? Looks like they are from Reference 74? This is the only time the authors mentioned these!
At the end of page 7 they wrote (Heavy metals such as nickel that are also found in PM have been theorized to inhibit RNA…) the breaking the news to the OBGYN doctors that Ni is not a heavy metal. It is a light first transition metal ion that is required for the activity of many metallo-proteins. If it is inhaled in WS sure it will have a devastating effect. Just being scientifically correct, Lead (Pb) and Mercury (Hg) are the traditional heavy toxic metals but not iron (Fe), nor Cobalt (Co), nor Nickel (Ni). We understand that Physicians will look at all metals as toxic/noxious if they are part of the WS.
In the same page they wrote: (In a study of 199 healthy pregnant women in Italy, PM10 exposure was associated with increased maternal mitochondria DNA) is the number 10 in the PM10 naming has to be subscripted? Such (PM10) as shown through the review! Similar to PM2.5.
Pages 8-13, all are good except on page 8, they wrote (as described by the Barker hypothesis [94].). Should not it read (as described by the Barker’s hypothesis [94]?)
Under page 11 they wrote (Exposure reduction strategies and implications for research public health authorities such as the U.S. Center for Disease Control and Prevention (CDC), U.S. Environmental Protection Agency (EPA),)…They abbreviated these well-known agencies (CDC & EPA) but never use them again. They defined EPA here late in page 11 but used it initially in page 6! They need to reverse this order (Define EPA in page 6 then reuse the EPA abbreviation again in page 11, because you have defined it above in page 6).
Page 14 under Figure 2 the chemical formula has to be written chemically correct: the authors wrote: nitrogen dioxide (NO2), Nitrogen Dioxide has the formula NO2 not NO2. With a subscripted 2.
Page 15-23 all are good.
Good work overall.
The abbreviation “Polycyclic aromatic hydrocarbons (PAHs)” needs to be included in the list of abbreviations.
References
References are good.
Under Supplementary file: The authors missed these tiny spelling words:
In the supplementary file there few typos/mistakes in the Tables provided: Table 1, Reference 4 7th column, they wrote: (temporally and spacially resolved gridded estimates of surface) the word spacially needs to be changed to specially.
In the same table, of supplementary reference 19 last column the word (amnio acids) needs to be changed to amino acids.
Best of Regards,

Round 2
Reviewer 1 Report
I should have been clearer in my initial comments on this paper, so my apologies. I have some major concerns with this paper, highlighted below.
This is a narrative review at best and should be described as such. There are no methods or description of how their review was conducted. As such, I have concerns over bias in their interpretation of the literature. I’m not sure what the journal policy is on review papers, but this is in no way a systematic review of this body of literature. There is no description of strengths and weaknesses of the literature that was reviewed.
A comprehensive review of wildfire smoke and birth outcomes was recently published in Environment International: Amjad et al. 2021, “Wildfire exposure during pregnancy and the risk of adverse birth outcomes: A systematic review”. This is an important review that is well structured and comprehensive. The current paper under review presents the same studies on birth outcomes and wildfires as the Amjad paper. Table 1 in both papers are essentially the same, with the exception of 1 new study in the current review by Heft-Neal (2022). The difference is that the Amjad paper was a comprehensive, systematic review with a description of methods and assessment of strengths/weaknesses and potential bias across studies. The current paper under review essentially presents no new information on wildfires and birth outcomes. The information they do present is highlighted more thoroughly by Amjad et al.
The current paper under review does present animal studies and human placental studies, but the vast majority of these are not related to wildfires but instead other types of air pollution exposures. This gets back at the comments in my initial review of the paper. This review of literature lacks focus and structure. Particularly for the animal and placental studies, the papers are not related to wildfires. I do not believe the authors are intentionally being misleading, but in my opinion there is simply not literature to back up the conclusions that wildfire-specific exposures impact these outcomes.
Given these points, this paper does not add anything to wildfire literature beyond what Amjad et al. have already published in a much more structured and comprehensive manner. I still cannot recommend this paper for publication and at this point leave it up to the journal editor.
